# Cartilage Laser Engraving for Fast-Track Tissue Engineering of Auricular Grafts

**DOI:** 10.3390/ijms252111538

**Published:** 2024-10-27

**Authors:** Anastas A. Kisel, Vladimir A. Stepanov, Elena V. Isaeva, Grigory A. Demyashkin, Evgeny I. Isaev, Ekaterina I. Smirnova, Elena M. Yatsenko, Grigoriy V. Afonin, Sergey A. Ivanov, Dmitrii A. Atiakshin, Petr V. Shegay, Andrey D. Kaprin, Ilya D. Klabukov, Denis S. Baranovskii

**Affiliations:** 1National Medical Research Radiological Center, Koroleva St. 4, 249036 Obninsk, Russia; ki7el@mail.ru (A.A.K.); dr.dga@mail.ru (G.A.D.); ekaterinasmirnova2702@gmail.com (E.I.S.); yatsenko@mrrc.obninsk.ru (E.M.Y.); dr.g.afonin@mail.ru (G.V.A.); oncourolog@gmail.com (S.A.I.); dr.shegai@mail.ru (P.V.S.); kaprin@mail.ru (A.D.K.); ilya.klabukov@gmail.com (I.D.K.);; 2Department of Laser and Plasma Technologies, Obninsk Institute for Nuclear Power Engineering, National Research Nuclear University MEPhI, Studgorodok 1, 249036 Obninsk, Russia; stepanov@iate.obninsk.ru (V.A.S.); e.isaev87@gmail.com (E.I.I.); 3Laboratory of Histology and Immunohistochemistry, Sechenov First Moscow State Medical University of the Ministry of Health of the Russian Federation (Sechenov University), 119991 Moscow, Russia; 4Research and Educational Resource Center for Immunophenotyping, Digital Spatial Profiling and Ultrastructural Analysis Innovative Technologies, Patrice Lumumba Peoples Friendship University of Russia (RUDN University), Mikluho-Maklaya St., 6, 117198 Moscow, Russia; atyakshin-da@rudn.ru; 5Department of Biomedicine, University Hospital Basel, University of Basel, 4001 Basel, Switzerland

**Keywords:** ear cartilage, nasal chondrocytes, laser engraving, regenerative medicine, tissue engineering

## Abstract

In this study, the optimal engraving parameters were determined through the analysis of scanning electron microscopy (SEM) data, as follows: a laser power density of 5.5 × 10^5^ W/cm^2^, an irradiation rate of 0.1 mm/s, a well radius of 60 μm, a distance between well centers of 200 μm, and a number of passes for each well of 20. After 1 week of in vitro cultivation, chondrocytes were located on the surface of the scaffolds, in the sockets and lacunae of decellularized cartilage. When implanted into animals, both cellular and acellular scaffolds were able to support cartilage in-growth and complete regeneration of the defect without clear boundaries with normal tissue. Nevertheless, the scaffolds populated with cells exhibited superior biocompatibility and were not subject to rejection, in contrast to cell-free constructs.

## 1. Introduction

The regeneration of damaged cartilage tissue represents a significant medical and social challenge. Cartilage deformation can result from a range of factors, including mechanical injuries, congenital anomalies, the natural aging process, and surgical interventions.

The natural regenerative capacity of cartilage is limited; the tissue is poorly saturated by cells with a low metabolic rate embedded in a dense and avascular extracellular matrix (ECM). Thus, cartilage defects commonly regenerate with fibrous connective tissue or fibrocartilage, which lack proper functional properties [1]. The continuous search for new options in cartilage regeneration in previous decades has resulted in the development of various clinical approaches. Previously, autoplastic surgeries were commonly used to restore the shape and function of damaged or congenitally deformed ear cartilage tissue [2]. However, with the advancements in regenerative medicine, tissue-engineered grafts have emerged as a promising solution for these purposes [3,4,5]. These tissue-engineered grafts primarily aim to create a suitable microenvironment for cells by keeping them close together, providing proper hydration, and stimulating their proliferation and metabolic activity. Overall, cartilage tissue engineering offers new possibilities for cartilage regeneration, addressing the limitations of routine surgical procedures and providing a more effective approach to repair [6,7].

Native scaffolds derived from decellularized extracellular matrices (dECMs) are preferred due to their ability to retain the original biochemical composition, microenvironmental architecture, and mechanical properties of the tissues from which they are derived [8,9,10,11,12,13,14,15].

Laser application is a standard technique in numerous medical fields, including head and neck surgery. In addition to tumor removal, lasers are employed to reshape cartilage and modify the surface of titanium dental implants with the objective of improving their biocompatibility [1,16,17,18]. Laser-induced microgrooving has the potential to enhance the functionality of cemented orthopedic implants [19]. In addition to doping with metal ions (Mg and Zn), laser patterning is employed to enhance the antimicrobial properties of ceramic surfaces utilized in the medical field [20]. Surface laser modification is employed in the synthesis of nanoparticles [21]. Flexible laser-scribed graphene (LSG) substrates with gold nanoislands have been developed as biochips for the in situ detection of biomolecules and viral proteins [22]. In recent years, laser radiation has emerged as a subject of considerable interest within the field of tissue engineering, as evidenced by a number of research studies in this area [23,24,25]. A laser-modified decellularized matrix has demonstrated favorable outcomes in dermal, nasal cartilage, and tubular tissue constructs, including small intestinal submucosa and trachea [26,27,28,29,30]. The surface of bioimplants may be ablated by laser radiation in a number of ways. Laser radiation can create micropores on the surface of various biomaterials without destroying them, thereby significantly increasing the contact area [31]. This technology is referred to as the laser microporous technique (LMT) [26,31] or laser micropatterning [32].

The laser engraving of cartilage with the formation of non-through holes also increases the specific surface area of the scaffold and stimulates the migration of seeded cells [28,33]. Laser-perforated scaffolds made of decellularized cadaveric cartilage and populated with nasal or auricular chondrocytes are capable of repairing tracheal [26,34,35], and knee-joint cartilage defects [36].

Laser matrix engraving by applying cross lines on the surface of human articular cartilage has been described by Nürnberger S. et al. [37]. This method allowed for the most regular and widespread distribution of cells, and the biomechanical properties of this scaffold were better than those of commercially available materials. In the event that the holes traverse the entirety of the specimen’s depth, this methodology may be classified as laser drilling [38]. The mechanical properties of menisci processed in this way remained within the physiological range. Before or after laser engraving, the cartilage is decellularized; engraving can be performed on fresh [31] or lyophilized cartilage samples [32,38].

In our study, we aimed to investigate the ability of a tissue-engineered grafts based on the perforated decellularized cadaveric cartilage of the rabbit auricle, populated with allogeneic nasal chondrocytes, to restore the defect of the external ear.

## 2. Results

### 2.1. Laser Perforation (Engraving) of Cartilage Tissue

The results of previous studies [33,34] were used as a basis for selecting the mode of laser treatment. Briefly, we found the following optimal parameters for engraving: laser power density—5.5 × 10^5^ W/cm^2^, irradiation speed—0.1 mm/s, radius of the wells—60 μm, distance between the well centers—200 μm, number of passes for each well—10.

We kept the optimal parameters for the laser engraving of the elastic cartilage but increased the number of passes up to 20. Either further increasing the number of passes (from 20 to 30) or enhancing the laser power (up to 6 × 10^5^ W/cm^2^), we observed undesirable effects in the form of carbonization and concave areas on cartilage samples (Figure 1I).

The cartilage engraving depth was measured. The depth and interquartile range (IQR) was 264.612 ± 411.831 µm with a median of 317.649 µm (Figure 1II). This distribution can be characterized by the non-uniform thickness of the taken cadaveric samples of cartilage tissue, as well as the passage of the laser through a complex heterogeneous system consisting of solid matrix and interstitial fluid [33]. The resulting wells exhibited a cone-shaped inlet diameter of 144.0 ± 22.5 μm, which subsequently decreased to 120 μm (Figure 1III,IV).

### 2.2. Isolation of Cells from Nasal Cartilages

During the primary isolation of cells from the nasal cartilages of rabbits, the number of cells was approximately 16.2 million, with a content of living cells of 94.1%. The cells designated as zero passage cells exhibited a cubic shape and granular cytoplasm, which is characteristic of chondrocytes (Figure 2I). These cells formed a dense monolayer on day 7 of cultivation. Alcian blue staining revealed specific metabolites of chondrocytes, GAGs, in the intercellular space (Figure 2II). This observation provided the basis for identifying isolated cells as cartilaginous.

### 2.3. Creation of Tissue-Engineered Grafts

Histologic sections of LPECs after 7 days of in vitro cell culture on scaffolds prior to animal implantation are shown in Figure 3. The sections show decellularized cartilage with laser-induced wells. Empty lacunae without cells are clearly visible. The intense red Safranin-O staining of the extracellular matrix demonstrates large amounts of GAGs in decellularized cartilage tissue. Cells are located on the upper surface of the scaffold and form a uniform layer invading the pores (Figure 3I–III). Single cells were present within the lacunae of the decellularized cartilage. The intercellular matrix in the cell-seeded layer was stained pink with Safranin O, indicating the synthesis of GAGs by adherent chondrocytes (Figure 3IV–VI). Less intense coloring indicates that the GAGs content in the intercellular matrix is low, which is explained by the limited time of scaffold cultivation. Thus, within 7 days, nasal chondrocytes seeded on the surface of the perforated scaffolds formed a layer of viable cells expressing cartilage-specific metabolites. Acellular LPECs are presented in Figure 4.

### 2.4. Animal Study

Tissue-engineered grafts were implanted into rabbits with full-thickness defects of the elastic cartilage in the external ear (Figure 5I–III). Each rabbit received the following two grafts: the scaffold seeded with cells was implanted in one ear, and the empty non-seeded scaffold without cells was implanted in the other ear. Throughout the observation period (36 days), the overall condition of the animals remained satisfactory. Wound healing was accompanied by mild inflammation, but in most cases was uneventful (Figure 5IV,V). Signs of rejection of the acellular implants were observed in three animals, which may be explained by the poorer viability of the empty scaffolds (Figure 5VI).

The histologic examination of the area of scaffold implantation revealed the integration of the tissue-engineered grafts into cartilage defects. There was no clear boundary between the scaffold and normal tissue. In addition, we observed the growth of newly formed cartilage tissue adjacent to both acellular (Figure 6(AI,AII)) and cellular (Figure 6(AIII,AIV)) constructs. The territorial matrix was intensely stained with Safranin O, resulting in a red-brown color (Figure 6(BI–BIV)). Immunohistochemical staining revealed a high content of type II collagen in the cytoplasm of chondroblasts (Figure 6(CI–CIV)). The cells were found not only in the newly formed cartilage tissue, but also in the cavities of the decellularized framework.

The cartilage consisted of sparsely distributed chondroblasts with outgrowths, typically single and not forming isogenic groups (Figure 7A). Upon the rejection of the implants the presence of multinucleated resorption cells, the presence of an inflammatory infiltrate containing a large number of eosinophilic leukocytes was noted (Figure 7(BI)). On the skin surface, a scab was detected, delimited from the necrotized tissues by a leukocyte shaft consisting mainly of neutrophils. Tissue-engineered grafts that were not integrated into normal tissue were subjected to resorption by macrophages and multinucleated cells of foreign body resorption (Figure 7(BII,BIII)).

## 3. Discussion

Suitable scaffold material is essential to succeed in auricular cartilage tissue engineering. The optimal scaffold should provide a three-dimensional environment to preserve the chondrocyte phenotype, as well as a porous structure and a surface suitable for cellular attachment [36,39]. Differentiation and proliferation occur in this artificial cell niche [40]. Tissue-engineered grafts should then have good mechanical properties to maintain the predetermined 3D shape for a long time after the implantation [41]. The time of scaffold biodegradation must match the rate of cartilage neogenesis [40]. Decellularized cartilage combines all features of the ideal scaffold for cartilage repair. However, the main disadvantage of this material is the relatively dense extracellular matrix, which resists the migration of chondrocytes and impedes the decellularization [31]. The laser surface modification of decellularized cartilage is used both to increase its surface area for adhesion and accelerate cell migration [16,28]. It also facilitates the decellularization process [26,31].

An appropriate wavelength, laser power and other settings are the most challenging part of applying a laser for tissue engineering. Tissue damage by overheating or even burning negatively influences the cell repopulation of the grafts. For example, in the study by Goldberg-Bockhorn E. et al. [28], the authors reported matrix compaction near perforations due to thermal effects, which prevented cell migration into tissue-engineered grafts from decellularized cartilage. This was observed to a greater extent when a CO_2_ laser was used. The pulsed laser irradiation has a smaller thermal effect and evaporates the tissue without the denaturation of collagen in the underlying and surrounding matrix. In our study, surface treatment with an infrared laser did not inhibit cell migration into the scaffold. Cells were observed both on the surface of the scaffolds and in the wells. Individual cells migrated into the collagen matrix within 7 days of in vitro culture, but it takes several weeks to fully repopulate the scaffold [28]. Histologic sections of scaffolds extracted 5 weeks after implantation show that they were enriched with cells. However, the origin of these cells needs to be further distinguished between migrated transplanted nasal chondrocytes and the host’s own cells, as cells were found in both acellular and chondrocyte-populated constructs. Importantly, the scaffolds in both cases stimulated cartilage growth de novo covering the implantation area.

Currently, there are several hypotheses about the mechanisms of ECM-promoted chondrogenesis in vitro and in vivo. The matrix itself is potentially chondroinductive due to the content of GAGs, such as chondroitin sulfate and aggrecan [42,43], collagen, and growth factors, especially from the transforming growth factor beta (TGF-β) superfamily [44,45]. The presence of type II collagen in the extracellular matrix is known to be beneficial for chondrogenesis [46,47]. Cramer M.C. and Badylak S.F. attributed the mechanical support, degradation and release of bioactive molecules, the recruitment and differentiation of endogenous stem cells (progenitor cells), and the modulation of the immune response to the anti-inflammatory phenotype to the mechanisms contributing to tissue remodeling [15]. In our previous study, we demonstrated that the incorporation of decellularized rib cartilage powder into collagen bioinks facilitates the formation of hyaline cartilage at the site of implantation of acellular tissue-engineered constructs and induces the differentiation of adipose-derived stem cells incorporated into the composition of the bioinks into chondrocytes [48]. The literature on tissue engineering contains relatively few studies where a laser was used to modify the surface of cartilage dECM. In one of them, the cartilage of the pig nasal septum was treated with a laser followed by its repopulation with human nasal chondrocytes [21]. As mentioned above, the efficiency of cell migration inside the scaffolds depended on the laser type and its parameters.

In their studies, Li Y. et al. [31] and Cheng J. et al. [49] used a CO_2_ laser to modify the surface of porcine articular cartilage. The authors reported that laser treatment increased the degree of decellularization of perforated cartilage tissue compared to intact cartilage and promoted cell adhesion. De novo cartilage in tissue-engineered constructs was formed both in vitro and in vivo when implanted into cartilage defects in the knee joints of animals.

In the previous work by Baranovskii D. et al. [34], as well as in the work of Xu Y. et al. [26], the cartilage tissue of the trachea of rabbits or humans was subjected to laser treatment. In the case of Xu Y. et al., the authors used a CO_2_ laser to treat the tissue-engineered grafts, as opposed to the infrared laser used in Baranovskii D.’s study. Subsequently, the grafts were revitalized with rabbit ear chondrocytes or human nasal chondrocytes, and implantation was performed subcutaneously in nude mice and in tracheal wall defects in rabbits. The effect after implantation was identical; the authors observed the formation of mature tubular cartilage with a relatively homogeneous distribution of cells as in micropores, and on both surfaces, without signs of rejection or stenosis.

Thus, a positive effect of laser treatment of decellularized cartilage grafts on cell adhesion and migration and cartilage tissue formation was observed both in vitro and in vivo. In our study, tissue-engineered grafts of rabbit auricular cartilage tissue with a laser-modified surface were used for the cartilage repair of the same organ. In contrast to the abovementioned works, the observation period for the experimental animals was 36 days. However, this time was sufficient for the grafts to integrate into the defect area and for cartilage growth de novo to begin. The better engraftment of cell-populated constructs compared to cell-free constructs should be noted. It is conceivable that the presence of cells may facilitate the accelerated remodeling of implanted tissue-engineered scaffolds. We did not aim to compare perforated and non-perforated tissue-engineered grafts. Our study demonstrates the feasibility of laser-treated scaffolds for the successful cartilage repair of the auricular cartilage. It is necessary to continue the study with a longer follow-up to ensure that the de novo formed cartilage tissue is elastic.

To date, the gold standard for the reconstructive treatment of human outer ear deformities continues to be transplantation of autologous rib cartilage [2]. However, the main limitation of this method is significant damage to the donor site and the need for several surgeries. Our encouraging results may pave the way for a one-stage reconstruction of the auricle with tissue-engineered grafts. Decellularized cadaveric cartilage is a suitable source for scaffold preparation, while patients’ autologic nasal chondrocytes could be cultured for its further revitalization. Rapid proliferation makes NCs the most promising source for simultaneous cartilage reconstruction among the other cells. According to our results, tissue-engineering grafts could be matured within one week prior to surgical reconstruction and immediately applied for simultaneous treatment.

## 4. Conclusions

The application of tissue-engineered grafts based on scaffolds derived from decellularized tissue has recently attracted increased attention. However, the physiological relevance of scaffolds and the difficulties of effective cell seeding require novel approaches for scaffold modification. We investigated a novel approach for the laser modification of the scaffold surface and the effectiveness of tissue revitalization. In our study, we demonstrated that NCs, which are already used in the clinic to repair cartilage defects, efficiently colonize the pores of LPECs and produce cartilaginous matrices in vivo. During the experiment, the following optimal laser engraving parameters were identified: the wells’ radius is 60 μm, with a distance of 200 μm between the centers, a laser power density of 5.5 × 10^5^ W/cm^2^, a speed of 0.1 mm/s, and a total of 20 passes. In addition, this study provided evidence that LPECs revitalized with rabbit NCs supported the stable reconstruction of EC defects and the growth of neo-formed cartilaginous tissue.

## 5. Materials and Methods

All operations with laboratory animals were approved by the Local Bioethical Control Committee (protocol no. 1-N-00038 18 August 2023).

### 5.1. Harvest and Devitalization of Rabbit EC (Ear Cartilage)

Cartilage tissue samples for tissue-engineered grafts were taken from the auricles of rabbit cadavers (≥2 years old at the moment of scarification). The samples were decellularized using the well-described freeze-and-thaw method [50]. In summary, the samples were placed into cryovials that were immersed in liquid nitrogen for 15 min, after which they were thawed in a bead bath (Lab Armor Bead Bath 6 L, Shellab, Cornelius, OR, USA) at 37 °C for 30 min. This cycle was repeated five times. The completeness of decellularization was controlled by histological study after staining with hematoxylin and eosin.

### 5.2. Laser Perforation (Engraving) of Decellularized Ear Cartilage (dEC)

A 1.06 µm fiber infrared laser (Raylogic galvo C8, Raylogic, Wuhan, China) was used to irradiate the cartilage. The characteristics of the laser source are summarized in Table 1.

The laser power was set to 5.5 × 10^5^ W/cm^2^ and the irradiation rate was 0.1 mm/s. Laser vaporization formed blind wells 120 μm in diameter, and a distance between the centers of 200 μm. The number of passes for each well was 20, 23, 30, and 40, with a duration of −1.5 min for each pass. Engraved samples were incubated for 1 h with 70% ethanol (while stirring every 15 min), and then the samples were washed 4 times with sterile phosphate-buffered saline (PBS) (1 day per wash, with a change in PBS every day) while shaking on a vortex (Heidolph, Schwabach, Germany). Next, they were placed in 100 μg/mL collagenase type I solution (Gibco, Waltham, MA, USA) in phosphate-salt buffer (pH, 7.4; Invitrogen, Waltham, MA, USA) for 2 h at 37 °C. Washing was performed twice with PBS. Next, DNAase was treated at a concentration of 5000 units/mL for 4 h at 37 °C. (50–375 U/μL, Thermo Fisher Scientific, Waltham, MA, USA). Further, engrave cartilages were washed twice with PBS and stored in PBS until use. The perforation depth was analyzed via microscopy (Leica DMi1, Leica Microsystem, Wetzlar, Germany; Flexacam C1 camera, Leica Microsystem, Germany) and ImageJ 1.52a. A depth probability density plot was made in R version 4.1.2.

### 5.3. Isolation and Culture of Rabbit Nasal Chondrocytes (rNCs)

rNCs were harvested according to Chen W. et al. [51]. Allogeneic rNCs were isolated from the nasal septum biopsy of 2 adult male rabbits (3.2–3.5 kg) under general anesthesia. Briefly, cartilage was washed in 70% alcohol, cut into 3–4 mm thick fragments, washed twice with PBS, and placed in a 0.15% type I collagenase solution (Gibco, USA) in Dulbecco’s Modified Eagle Medium (DMEM), glucose content 4.5 g/k (Gibco, USA), then incubated at 37 °C and 5% CO_2_ for ~20 h in a CO_2_ incubator (SCO5W, Shellab, USA). After incubation for final cell separation, the solution was stirred for 30 min on a magnetic stirrer (Biosan, Riga, Latvia), then cells were precipitated for 5 min at 400 g (ELMI CM-6M centrifuge, Elmi, Riga, Latvia). For enzyme washing, the cell sediment was resuspended in DMEM (glucose content 4.5 g/L) (Gibco, USA) containing 10% fetal bovine serum (FBS) (Biosera, Cholet, France) and centrifuged in the same mode. The procedure was repeated twice.

Then, the cell suspension was transferred into culture vials (Corning, Corning, NY, USA) with a surface area of 75 cm^2^ at the rate of at least 10,000 cells per cm^2^. The isolated rNCs were cultured in DMEM containing 10% FBS, 10 mM HEPES (4-(2-hydroxyethyl)-1-piperazineethanesulfonic acid) buffer (Gibco, USA), 1 mM sodium pyruvate (Gibco, USA), 0.29 mg/mL l-glutamine, 100 U/mL penicillin, 100 µg/mL streptomycin (Gibco, USA) for 2 passages. The medium was refreshed twice a week.

Some cells were cultured in 3.5 cm Petri dishes (Corning, NY, USA, 300 × 10^3^ cells per dish) in order to identify chondroblasts that are capable of forming glycosaminoglycans (GAGs). The cells were cultured under standard conditions for seven days. Thereafter, the culture medium was removed, and the monolayer was washed twice with PBS. It was then fixed with 4% glutaraldehyde (PanReac, Castellar del Vallès, Spain) at room temperature. Following the application of 0.1 N HCl for the purpose of washing the cells, the monolayers were stained with a 1% alcian blue solution (8GX, Sigma-Aldrich, Saint Louis, MO, USA, in 0.1 N HCl) at room temperature. The monolayers were then washed twice with 0.1 N HCl, dried, and moistened again with 0.1 N HCl. The dishes were examined using a microscope (Leica DMi1, Leica Microsystem, Germany; Flexacam C1 camera, Leica Microsystem, Germany).

### 5.4. Revitalization of Laser-Perforated Ear Cartilages (LPECs)

LPECs were placed onto 0.4 mm pore-size polycarbonate Transwell filters (Corning B.V. Life Sciences, SchipholRijk, Amsterdam, The Netherlands; 2 samples/insert). rNCs were loaded on the top of the LPECs (100 μL of cell suspension containing 0.5 × 10^6^ cells per insert) in Chondrogenic Medium, high-glucose DMEM containing 5% FBS, 10 mM HEPES, 1 mM sodium pyruvate, 100 U/mL penicillin, 100 µg/mL streptomycin, 0.29 mg/mL l-glutamine, 10 µg/mL insulin (Novo Nordisk, Bagsvaerd, Denmark), and 0.1 mM ascorbic acid 2-phosphate (Sigma-Aldrich) for 7 days at 37 °C and 5% CO_2_ with media changes twice/week. Acellular and revitalized LPEC grafts with rNCs were assessed histologically. Acellular and revitalized LPEC grafts with rNCs were implanted orthotopically in rabbits.

### 5.5. In Vivo Experiments: Orthotopic Study in Rabbits

Six-month-old male rabbits (n = 5) of the breed “Gray Giant”, weighing ~3 kg, were used as experimental animals. Cartilage defects were applied to the right and left ear of each animal. To create the defect, an incision was made on the auricle, and the rabbit’s own cartilage was cut out 5 × 5 mm in size and full depth. Then, a perforated cartilage populated with cells or without cells was placed in the defect and fixed with suture material (Ethicon Prolene, Scotland). Surgical intervention was performed under general anesthesia. The operation on the implantation of tissue-engineered grafts was performed in the clinic of veterinary radiology at the A. Tsyb Medical Radiological Research Centre—Branch of the National Medical Research Radiological Centre of the Ministry of Health of Russia (A. Tsyb MRRC—a Branch of “NMRC Radiology” of the Ministry of Health of Russia). The following preparations were used for anesthesia and analgesia: medetomidin 0.2 mg/kg of live weight p/k; 4% isoflurane inhaled; and supplemented with lidocaine 20 mg/mL topically. During the experiment, rabbits were isolated from each other in single cages with free access to water and food. The operated areas were examined daily. After 36 days, the rabbits were taken out of the experiment with the subsequent removal of the implant with the surrounding tissues from the auricle.

### 5.6. Scanning Electron Microscopy

We used environmental scanning electron microscopy (SEM) in the saturated water vapor (Nova NanoSEM 230, FEI, Hillsboro, USA, Acc: 5000 Volt, Detektor: 8000, Iteration: 22,700 nA) to visualize the surface of the laser-engraved scaffold, measure pore diameter and distances between the pore centers. The samples were fixed overnight in 0.05% glutaraldehyde at 4 °C, dehydrated in graded ethanol concentrations, and processed via “critical point drying” (CPD).

### 5.7. Histology

Scaffolds before implantation and scaffolds with fragments of surrounding tissue extracted one month after implantation were fixed for 24 h in 10% buffered HistoSafe^®^ formalin (BioVitrum, Saint Petersburg, Russia). After washing in 70% ethanol, standard histologic wiring of specimens was performed, followed by encapsulation in paraffin medium (Histomix, BioVitrum, Russia). Paraffin sections of 5 μm thickness obtained on a microtome (Leica RM2235, Leica Biosystems, Nussloch, Germany) were placed on silanized slides (S3003, Dako, Santa Clara, CA, USA). For histologic studies, deparaffinized sections were stained with hematoxylin and eosin, and Safranin O/Fast Green Stain (Wuhan Servicebio Technology, Wuhan, China). After dehydration in alcohols and clarification in (ortho-)xylene, the preparations were encapsulated in Canada balsam (Merck, Darmstadt, Germany).

### 5.8. Immunohistochemistry

Immunohistochemical studies were performed using polyclonal mouse antibodies to collagen type II (II-II6B3 (Hybridoma Bank, Iowa City, IA, USA, 1:100). Secondary rabbit antibodies to mouse IgG conjugated to biotin (B8520-1ML, Sigma,(Merck KGaA), Darmstadt, Germany, 1:600) were used for the immunostaining of mouse antibodies. According to the immunohistochemical protocol, deparaffinized sections immersed in citrate buffer (pH 6.0) were boiled (5 min) before the application of primary antibodies to type II collagen. Endogenous peroxidase was blocked in 3% hydrogen peroxide solution. A total of 2% normal serum of secondary antibody donor animals, 1% bovine serum albumin and 0.1% Triton X-100 (AppliChem GmbH, Darmstadt, Germany) were added to the blocking buffer. The preparations were incubated in the primary antibody solution overnight in a humid chamber at +4 °C. After washing the preparations in PBS, secondary rabbit anti-mouse antibodies were applied to the slices for 30 min at room temperature. PBS was washed again and ExstrAvidin^®^-Peroxidase (E2886-1ML, Sigma, (Merck KGaA), Darmstadt, Germany, 1:250) was applied for 30 min at room temperature. Substrate peroxidase was detected using diaminobenzidine (Liquid DAB+, K3468, Dako, Santa Clara, CA, USA). After dehydration in alcohols and clarification in xylene, the preparations were encapsulated in Canada balsam. Histologic sections were studied under an AXIO Imager A1 microscope (Carl Zeiss, Oberkochen, Germany) with microphotography using a Canon Power Shot A640 digital (Canon, Tokyo, Japan) camera.

## Figures and Tables

**Figure 1 ijms-25-11538-f001:**
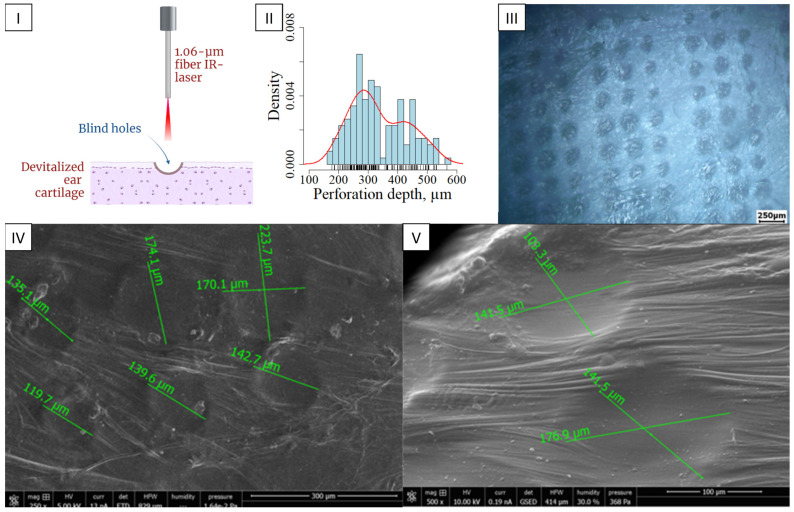
Laser engraving of the devitalized ear cartilage. (**I**) The principle of laser engraving of the devitalized ear cartilage. (**II**) Probability density of perforation depth, n = 132. The red line shows the probability density curve; (**III**) Light microscopy of dECM laser engraving, objective lens ×5, scale bar 250 μm. (**IV**) SEM laser-engraved dECM of rabbit auricle, scale bar 300 μm. (**V**) SEM laser-engraved dECM of rabbit auricle, scale bar 100 μm.

**Figure 2 ijms-25-11538-f002:**
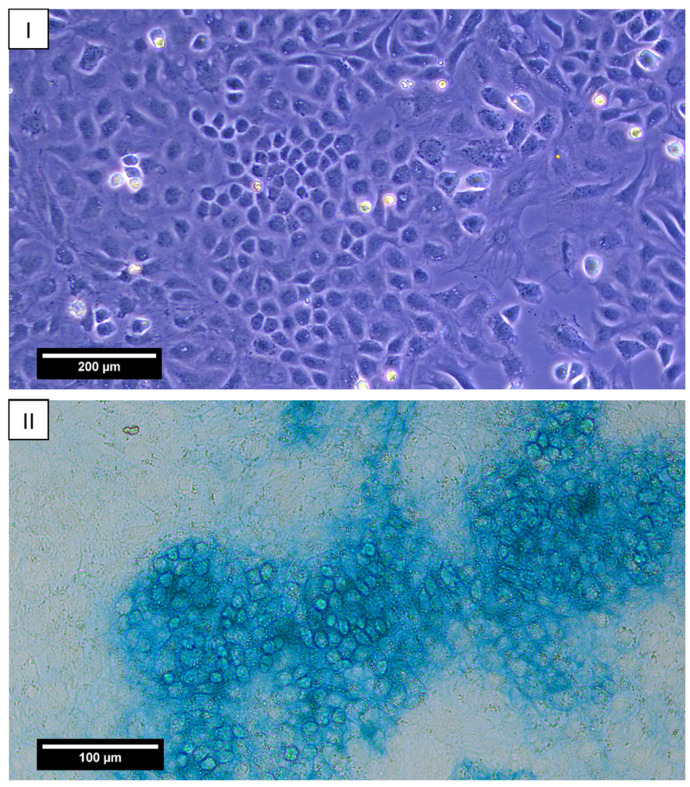
Chondrocytes of the 0th passage after 7 days of cultivation in a Petri dish. (**I**) Phase contrast, objective lens ×10, scale bar—200 μm. (**II**) Alcian blue staining, glycosaminoglycans. Objective lens ×20, scale bar—100 μm.

**Figure 3 ijms-25-11538-f003:**
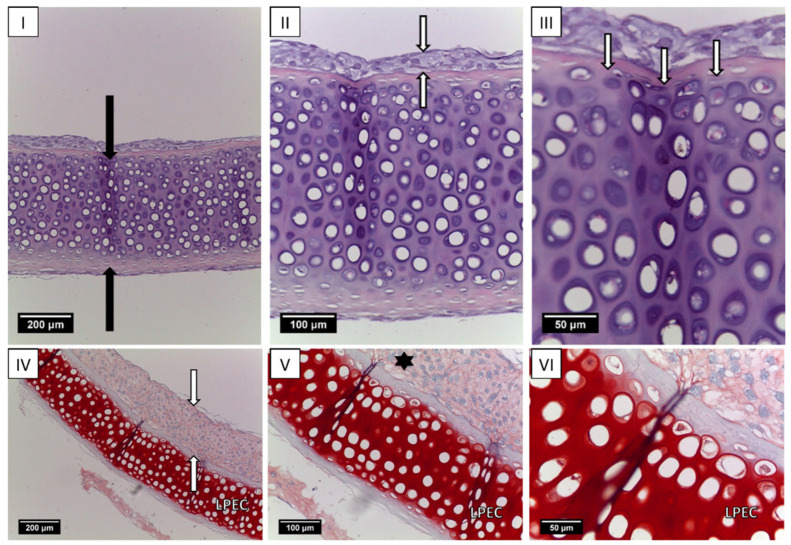
A cross-section of LPECs populated with nasal chondrocytes after 7 days of in vitro cultivation. Hematoxylin and eosin staining. (**I**) The black arrows indicate a formed non-square perforation, objective lens ×10, scale bar—200 μm. (**II**) The white arrows indicate the layer of attached nasal chondrocytes, objective lens ×20, scale bar—100 μm. (**III**) The white arrows show cells filling the perforation in LPECs, objective lens ×40, scale bar—50 μm. A longitudinal section of LPECs populated with nasal chondrocytes after 7 days of in vitro cultivation. Safranin-O staining. (**IV**) The white arrows indicate the formation of the nasal chondrocyte layer, objective lens ×10, scale bar—200 μm. (**V**) GAG production in the intercellular substance of repopulated chondrocytes (asterisk), objective lens ×20, scale bar—100 μm. (**VI**) Objective lens ×40, scale bar—50 μm.

**Figure 4 ijms-25-11538-f004:**
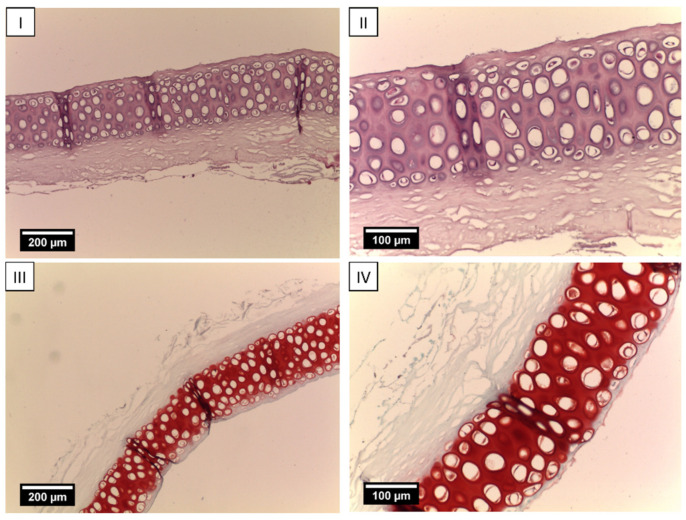
Cell-free LPECs. (**I**) A longitudinal section of LPECs, objective lens ×10, scale bar—200 μm. Hematoxylin and eosin staining. (**II**) Objective lens ×20, scale bar—100 μm. Safranin-O staining. (**III**) Objective lens ×10, scale bar—200 μm. Safranin-O staining. (**IV**) Objective lens ×20, scale bar—100 μm. Safranin-O staining.

**Figure 5 ijms-25-11538-f005:**
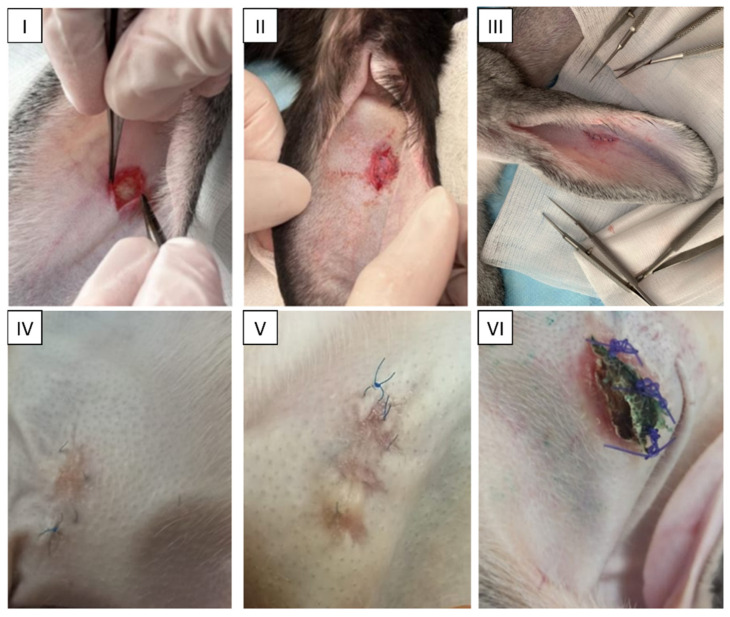
In vivo orthotopic study. LPEC transplant surgery. General view of the operated area. Creation of a cartilage defect (**I**), placement of LPEC in the defect (**II**) and the suturing of the operated area (**III**); general view of the surgical wound with nasal chondrocyte cells populated (**IV**) and control group (**V**) in rabbit auricles. Rejected graft of the control group in the rabbit’s auricles (**VI**).

**Figure 6 ijms-25-11538-f006:**
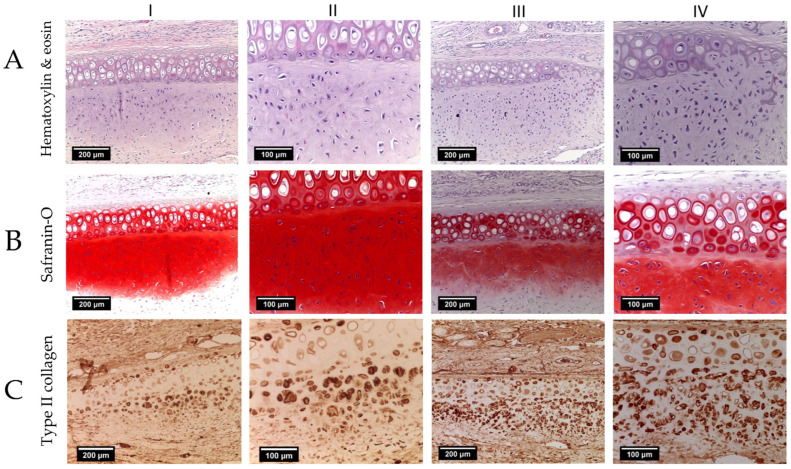
Transverse section of cartilage tissue of the control (**I**,**II**) and experimental (**III**,**IV**) groups. (**A**) Hematoxylin and eosin staining, (**B**) Safranin-O staining and (**C**) Immunohistochemical staining for type II collagen. Objective lens ×10, scale bar—200 μm (**I**,**III**) and objective lens ×20, scale bar—100 μm (**II**,**IV**).

**Figure 7 ijms-25-11538-f007:**
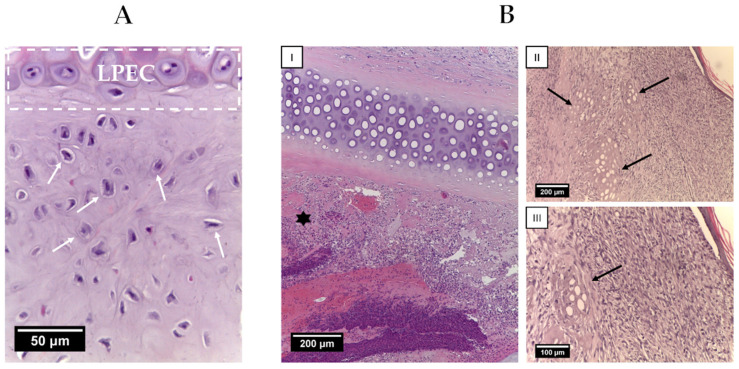
(**A**) An overgrowth of newly formed cartilage tissue. The dotted area indicates LPECs. White arrows indicate single chondroblasts. Hematoxylin and eosin staining. Objective lens ×40, scale bar—50 μm; (**B**) the inflammatory responses and resorption of LPECs. (**I**) Inflammatory infiltrates in the area of rejection LPECs. Cellular infiltrate (asterisk). Hematoxylin and eosin staining, objective lens ×10, scale bar—200 μm. (**II**) Resorption of implanted LPECs. The arrow indicates the remains of the structure. Hematoxylin and eosin staining, objective lens ×10, scale bar—200 μm and (**III**) Resorption of implanted LPECs. The arrow indicates the remains of the structure. Hematoxylin and eosin staining, objective lens ×20, scale bar—100 μm.

**Table 1 ijms-25-11538-t001:** The characteristics of the laser source.

Laser source type	ytterbium fiber laser source
Laser source power	20 W
Pulse frequency range	20–50 kHz
Laser source wavelength	1064 nm
Positioning accuracy	3 µm
Pulse energy	0.33–1 mJ
Contact spot	50 µm
Laser Source Cooling	Air
Type of lifting mechanism (Z-axis)	Manual drive on the scanning stand
Software	EZCAD 2.14

## Data Availability

Data is contained within the article. The original contributions presented in the study are included in the article, further inquiries can be directed to the corresponding author/s.

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
