# Peer review of "Cartilage Laser Engraving for Fast-Track Tissue Engineering of Auricular Grafts"

_ijms, 2024, doi:10.3390/ijms252111538_

Round 1

Reviewer 1 Report

Comments and Suggestions for Authors

This manuscript provides valuable insights into the use of laser-modified decellularized cartilage scaffolds for cartilage regeneration, demonstrating the benefits of surface modification in enhancing chondrocyte migration and tissue integration. The experimental design is thorough, with clear evidence supporting the improved biocompatibility of cell-populated scaffolds. In general, this manuscript has been well organized and written. However, there are several shortcomings which need to be improved before being accepted.

1.    Some words in Figures 1 (II) and (V) are too small and unclear, please enlarge them.

2.  The manuscript reports specific laser parameters used for scaffold modification but provides limited mechanistic discussion on how these parameters were optimized. Could the authors elaborate on the rationale for selecting these particular parameters, and explain how variations in these settings might influence the scaffold's structural and biological properties?

3.     Reference: There are some typographical errors in the references. Please review and correct them.

Reviewer 2 Report

Comments and Suggestions for Authors

Dear Authors,

The presented work is interesting and can be useful in different biomedical applications. However, the manuscript needs to be revised and improved before publication.

Please see my comments below:

- It seems that the term “Introduction:” is missing from the beginning of the Abstract.

- I think that the Abstract is long and needs to be shortened. 

- No keywords are included.

- manuscript can benefit from language editing and proofreading

- The introduction needs to be improved. At the beginning of the introduction section, please add some general information on the topic and the field of the proposed research for general readers. 

- In the introduction section, please elaborate on the application of laser radiation and surface laser modification (including laser engraving technologies) in medical applications. A brief literature review on existing methods and cutting-edge laser engraving technologies related to the scope of your research needs to be added to the manuscript.

- I think many figures are merged in Figure 1 that should be separated and included under the related paragraphs in the related section.

Round 2

Reviewer 2 Report

Comments and Suggestions for Authors

Dear Authors,

Thank you for reflecting the comments. 

The revised manuscript looks good.

Good Luck